# The Effects of Persimmon (*Diospyros kaki* L.f.) Oligosaccharides on Features of the Metabolic Syndrome in Zebrafish

**DOI:** 10.3390/nu14163249

**Published:** 2022-08-09

**Authors:** Wanlapa Nuankaew, Hyo Kyu Lee, Youn Hee Nam, Ji Heon Shim, Na Woo Kim, Sung Woo Shin, Min Cheol Kim, Seung Yeon Shin, Bin Na Hong, Sukanya Dej-adisai, Jong Hwan Kwak, Tong Ho Kang

**Affiliations:** 1Department of Oriental Medicine Biotechnology, College of Life Sciences, Kyung Hee University Global Campus, Yongin-si 17104, Korea; 2Department of Pharmacognosy and Pharmaceutical Botany, Faculty of Pharmaceutical Sciences, Prince of Songkla University, Hat Yai 90112, Thailand; 3School of Pharmacy, Sungkyunkwan University, Suwon-si 16419, Korea

**Keywords:** *Diospyros kaki*, persimmon, metabolic syndrome, type 2 diabetes, obesity, oligosaccharide

## Abstract

Metabolic syndrome has become a global health care problem since it is rapidly increasing worldwide. The search for alternative natural supplements may have potential benefits for obesity and diabetes patients. *Diospyros kaki* fruit extract and its oligosaccharides, including gentiobiose, melibiose, and raffinose, were examined for their anti-insulin resistance and obesity-preventing effect in zebrafish larvae. The results show that *D. kaki* oligosaccharides improved insulin resistance and high-fat-diet-induced obesity in zebrafish larvae, evidenced by enhanced β-cell recovery, decreased abdominal size, and reduced the lipid accumulation. The mechanism of the oligosaccharides, molecular docking, and enzyme activities of PTP1B were investigated. Three of the oligosaccharides had a binding interaction with the catalytic active sites of PTP1B, but did not show inhibitory effects in an enzyme assay. The catalytic residues of PTP1B were typically conserved and the cellular penetration of the cell membrane was necessary for the inhibitors. The results of the mechanism of action study indicate that *D. kaki* fruit extract and its oligosaccharides affected gene expression changes in inflammation- (*TNF-α*, *IL-6*, and *IL-1β*), lipogenesis- (*SREBF1* and *FASN*), and lipid-lowering (*CPT1A*)-related genes. Therefore, *D. kaki* fruit extract and its oligosaccharides may have a great potential for applications in metabolic syndrome drug development and dietary supplements.

## 1. Introduction

Recently, metabolic syndrome has become a common occurrence in the world, with a prevalence of 10–40% [1]. The consumption of quality fats and carbohydrates is pivotal to improve blood lipids and glycemic index in metabolic syndrome patients [2]. Type 2 diabetes is related to hyperglycemia and insulin resistance, which are mainly caused by obesity [3]. Insulin resistance is the early stage of type 2 diabetes, when high glucose blood levels lead to an increase in pancreatic insulin production [4], which results in increased glucose output and lipogenesis via the liver [5].

There is considerable evidence showing that the intracellular protein PTP1B (Protein Tyrosine Phosphatase 1B) is upregulated in muscle and adipose tissues of obese and insulin-resistant mice [6]. The inhibition of PTP1B improved energy expenditure, reduced fat deposition, and prevented weight gain in obese mice [7]. PTP1B is able to impair the glucose uptake by interacting with IR and IRS-1 to induce the tyrosine phosphorylation hydrolysis of insulin activity [8].

Using zebrafish (*Danio rerio*) as an animal model, as it has a rapid reproduction and morphology development, allows drug screening in a short period of time, lower maintenance costs, and small amount of test samples. Moreover, zebrafish share 84% similarity to mammalian disease genes [9]. The similarity of the zebrafish pancreas structure to that of mammals [10], of normal blood glucose levels (human: 70–120 mg/dL; adult zebrafish: 50–75 mg/dL) [11], and adipocytes in larvae make zebrafish a potential tool to investigate metabolic syndrome [12]. Several previous studies used zebrafish as a model of metabolic syndromes, such as Moss et al. [13] and Intine et al. [14], who induced diabetes in zebrafish with streptozotocin and observed the regeneration of the pancreas in adult zebrafish.

*Diospyros kaki* (persimmon) fruit is known as a valuable nutrient source. The dried pulp of *D. kaki* decreased the plasma glucose levels and triglycerides of and inhibited digestive enzymes in rats [15]. Moreover, *D. kaki* leaf has been reported to lower blood glucose levels in a rodent model by inhibiting the pancreas alpha-amylase enzyme [16]. The triterpenoids from their leaves showed an inhibitory activity of PTP1B [17] and the proanthocyanidins from the peel had an effect on α-glucosidase or α-amylase enzyme [18].

Oligosaccharides are natural products that have gained attention in the treatment of metabolic syndrome [19]. Because the glycosidic bond between the monomeric sugar units of oligosaccharides cannot be broken down by human gastrointestinal digestive enzymes, they are nondigestible in the human digestive system [20]. Previous research showed several beneficial effects of functional oligosaccharides, such as improving pancreas function, insulin and leptin resistance, inflammation, inhibiting α-glucosidase, and regulating gut microbiota and hormones, though largely the mechanism still remains unclear. Thus, studies on the mode of action of oligosaccharides are beneficial to metabolic syndrome drug development [19].

## 2. Materials and Methods

### 2.1. Preparation and Extraction of D. kaki Fruit Extract and Source of the Oligosaccharides

*D. kaki* fruits were purchased from the market in Suwon, South Korea. Fruit flesh was cut into small pieces. A total of 100 g of the flesh and 100 mL of distilled water were blended using a food processor and then filtered. The juice of *D. kaki* fruit was evaporated by Smart Evaporator S Spiral Plug (BioChromato, Kanagawa-ken, Japan) at the temperature of 55 °C until completely dried to yield 7.64% based on the fresh weight. Gentiobiose, melibiose, and raffinose were purchased from Tokyo Chemical Industry Co., Ltd. (Tokyo, Japan).

### 2.2. The Investigation of the Chemical Constituent of D. kaki Fruit Extract

The chemical constituent analysis of *D. kaki* was performed using ultra-high performance liquid chromatograph-mass spectrometer/mass spectrometer 1290 Infinity II UHPLC-6545 Q-TOF (Agilent Technologies, Santa Clara, CA, USA) with an Agilent EclipsePlus C_18_ column (2.1 mm × 50 mm i.d., 1.8 μm; flow rate: 0.2 mL/min) maintained at 35 °C and the injection volume was 10 µL. To optimize the separation of the sample, two types of mobile phase included 0.1% formic acid–water (A) and acetonitrile (B). The optimized linear gradient elution was performed as follows: 0–20 min, 18% B; 21–35 min, 20% B; and 36–70 min, 30% B. Mass spectroscopy was performed on quadruple time-of-flight mass spectrometer (Q-TOF MS) and characterization equipped with electrospray ionization Dual AJS ESI. Mass spectrometry data were acquired in the negative and positive ionization modes covering the range from *m*/*z* 100 to *m*/*z* 1500 with scan rate of 2 spectra/s, nebulization with nitrogen at 35 psi, dry gas flow of 13 L/min at a temperature of 325 °C, and skimmer 1 voltage 65 V, and collision energy was set to 10%, 20%, and 40%. The instrument was operated and analyzed using Agilent Technologies. Mass spectra were analyzed by MS/MS spectral mass library.

### 2.3. Animals

Zebrafish (*Danio rerio*) were maintained in S type system (1500 × 400 × 2050 mm, Woojung Bio, Inc., Suwon, Korea) at 28.5 °C under a 10/14 dark–light cycle. Three pairs of adult zebrafish were breeding overnight, and the embryos were collected three hours post-fertilization (hpf). The embryos were washed and renewed with 0.03% sea salt solution or treatment solution every day until the experiment was performed. The zebrafish maintenance was followed standard protocols approved by the Animal Care and Use Committee of Kyung Hee University (KHUASP[SE]-15-10).

### 2.4. Anti-Insulin Resistance in Zebrafish

Three-day post-fertilization (dpf) wild-type zebrafish larvae (20 larvae/group) were used for the experiments. A total of 10 µM of the biosynthetic human insulin solution (Novo Nordisk, Kalundborg, Denmark) was used for degenerating the pancreatic islet (PI) of the zebrafish larvae. After 48 h of induction, the larvae were rinsed with 0.03% sea salt solution and treated by 10 µg/mL of fruit extracts (contained approximate amounts of gentiobiose, melibiose, and raffinose at 11.7, 7, 1.3 ng/mL, respectively) and 1 µM of pure compounds (342.3 ng/mL of gentiobiose, 342.3 ng/mL of melibiose, and 504.4 ng/mL of raffinose) for another 48 h.

Finally, the larvae were stained with 40 µM 2-NBDG (Invitrogen, Life Technologies, Grand Island, NY, USA) for 30 min and imaging PI picture by a fluorescence microscope (Olympus, Tokyo, Japan). PI size was measured and analyzed by Focus Lite software (Focus Co., Daejeon, Korea) to monitor β-cell ablation [21].

### 2.5. Obesity-Preventing Effect in Zebrafish

A total of 5 dpf of wide-type zebrafish were placed in 20 mL of 0.03% sea salt solution (20 larvae per group). The normal group (NOR) was fed by 4 mg of commercial standard diet (ACE KOREA, Korea), and the obesity group (OBS) was fed by 20 mg of high-fat diet (hard-boiled egg yolk). A total of 10 µg/mL of *D. kaki* fruit extract and 1 µM of the oligosaccharides were provided when start feeding induction to explore the obesity-preventing effect. After three days of feeding, all groups were starved and the solution was renewed. The treatment groups remained with the *D. kaki* fruit extract and oligosaccharide maintenance. The fasting and treatment took another two days (the experimental setup is shown in Figure 1A). To monitor initial adiposity, larvae belly length was measured and imaged the abdominal part of larvae using 0.5 µg/mL of Nile red staining and fluorescence microscopy on the fifth day of the experiment (image shown in Figure 1B).

### 2.6. Molecular Docking of Isolated Compounds and PTP1B

The analysis of molecular binding interactions of oligosaccharide structures with PTP1B enzyme were performed using AutoDock Vina version 1.1.2. The structure of PTP1B (PDB code: 1T49) was downloaded from RCSB Protein Data Bank. The 3D structure of gentiobiose (CID: 441422), melibiose (CID: 440658), and raffinose (CID: 439242) were obtained from PubChem Compound (NCBI). Discovery Studio 2019 was used for removing ligands and water molecules. Hydrogen atoms and charges were then added using the AutoDock Tool. The grid maps were generated with a default spacing of 0.375 Å and 50 × 50 × 50 grid box size. The X, Y, Z grid maps centers were 51.013, 27.372, and 22.603, respectively. All of the molecular bonds of the isolated compounds were set to rotatable. All torsions were also allowed to rotate. The interactions of molecular docking were analyzed using Discovery Studio 2019. The best affinity of the molecular interaction was determined by the lowest binding energy of hydrogen bond between the active residues and compounds [22].

### 2.7. Enzyme Activity Assays of PTP1B

A total of 1 mM EDTA (Sigma-Aldrich Co., St. Louis, MO, USA), 0.15 M NaCl (Sigma-Aldrich Co., St. Louis, MO, USA), and 3 units/mL of human-recombinant PTP1B (BioVision, Inc., Milpitas, CA, USA) were dissolved in sodium acetate buffer (pH 5.5) to prepare the reaction mixture. Suramin (Sigma-Aldrich Co., St. Louis, MO, USA) was used as a positive control. A total of 100 µL of the samples were mixed with 50 µL of reaction mixture and incubated at 30 °C for 10 min. A total of 50 µL of the substrate 10 mM ρ-nitrophenyl phosphate (PNPP) (Sigma-Aldrich Co., St. Louis, MO, USA) was added and incubated at 30 °C for 20 min. Then, 100 mM of NaHCO_3_ (Sigma-Aldrich Co., St. Louis, MO, USA) was added to stop the reaction and the absorbance was measured at 415 nm [23].

### 2.8. Real Time-Quantitative Polymerase Chain Reaction (RT-qPCR)

Zebrafish larvae were fed with a high-fat diet for three days and then treated with the *D. kaki* fruit extract and oligosaccharides for another two days. The extraction of total RNA was performed using TRIzol^TM^ reagent (Thermo Fisher Scientific, Seoul, Korea). Reverse transcription was performed by Reverse Aid First Strand cDNA Synthesis Kit (Thermo Fisher Scientific, Seoul, Korea) for the synthesis of cDNA. The qPCR was performed in 10 μL reactions containing 5 μL of SYBR Select Master Mix (Applied Biosystems, Waltham, MA, USA, Thermo Fisher Scientific), 1 μL of cDNA template, 1 μL of forward primer (10 pmol), 1 μL of reverse primer (10 pmol), and 2 μL of RNase free water. The qPCR parameters were initial denaturation at 95 °C for 5 min, followed by 45 cycles of 95 °C for 15 s, 60 °C for 15 s and 72 °C for 20 s, and then 73 °C for 5 min. Primer sequences are listed in Table 1. The expression of genes was analyzed by the 2^−ΔΔCt^ method.

### 2.9. Statistical Analysis

The result data were expressed as mean with standard error of the mean (±SEM). A one-way ANOVA with Dunnett’s post hoc test of GraphPad Prism version 8 was used to verifying the statistical significance. A *p* value < 0.05 (*) was considered statistically significant.

## 3. Results and Discussions

### 3.1. Screening of D. kaki Fruits Extracts for Anti-Insulin Resistance in Zebrafish

The investigation of anti-insulin resistance in larvae zebrafish was established to screen the natural products for the treatment of metabolic syndrome. A high dose of human insulin (10 µM) was provided to stimulated insulin resistance in zebrafish larvae by downregulating the immune system and insulin signaling pathway. The PI size of zebrafish larvae were measured to evaluate the β-cell existence [20]. The human insulin significantly lowered the size of the larvae PI compared to the normal group (*p* < 0.0001), while the treatment with 10 µg/mL of *D. kaki* fruit extract significantly increased the PI size compared to the insulin-induced group (*p* < 0.01) (Figure 2). Thus, the fruit extract of *D. kaki* was considered for the further investigation of anti-metabolic syndrome activities by their enhancing β-cell subsistence in insulin-resistant larvae zebrafish.

### 3.2. Chemical Constituent Analysis by HPLC-MS/MS

The chemical constituents of the *D. kaki* fruit extract were analyzed using HPLC-MS/MS via ESI(−) (Figure 3A) and ESI(+) (Figure 3C) modes. A total of 10 compounds were detected in ESI(−) and 28 compounds were detected in ESI(+). The compounds’ name, retention times, peak high, and mass of the compounds are shown in Table 2. Various classes of natural compounds were found, especially several types of oligosaccharides, including, raffinose, gentiobiose, neokestose, melibiose, and sophorotriose (the approximate percent contents are 0.13, 1.17, 1.21, 0.70, and 0.72, respectively). Mass spectrum of the oligosaccharides found in *D. kaki* fruit extract were shown in Figure 4. According to the non-digestible and probiotic features of functional oligosaccharides, such as galacto-oligosaccharides, xylo-oligosaccharides, and fructo-oligosaccharides. Their anti-diabetic effect has been reported [19]. However, limitations in the knowledge of oligosaccharides’ structure and their efficiency remain. Therefore, the study of their mechanism in metabolic syndrome is required to evaluate the ability of oligosaccharides.

### 3.3. Molecular Docking of PTP1B to the Oligosaccharide from D. kaki Fruit

To understand the mechanism of the selected oligosaccharides with PTP1B, molecular docking studies were performed. The binding interaction of oligosaccharides’ structure and PTP1B was performed via AutoDock Vina 1.1.2. The active site of PTP1B includes the allosteric site and the catalytic domain, which consist of the catalytic site, catalytic loop, and catalytic WPD loop [24]. The results of in silico molecular docking show that the hydroxyl groups of gentiobiose binding to Pro180 residue of the catalytic WPD loop, Cys215 of the catalytic domain, and Ala217 of the catalytic loop with the hydrogen bond interaction and binding energy of −6.9 kcal/mol. The O-glycosidic bond of melibiose binding with Cys215 of the catalytic domain and O in the backbone bound to the Arg221 residue of the catalytic loop with a binding energy of −7.3 kcal/mol. The hydroxyl groups of raffinose were bound to the Pro180 and Cys215 amino acid residues of the catalytic WPD loop and catalytic domain, respectively. The affinity of the binding reaction is −7.1 kcal/mol. Neokestose and sophorotriose did not show a hydrogen bond interaction with the active residues of PTP1B. All of the results are shown in Figure 5. Therefore, the oligosaccharides gentiobiose, melibiose, and raffinose were selected for further experiments.

### 3.4. Anti-Insulin Resistance in Zebrafish Induced by Oligosaccharides from D. kaki Fruit

The oligosaccharides gentiobiose, melibiose, and raffinose were investigated against insulin resistance in larvae zebrafish with a concentration of 1 µM. The results show that gentiobiose, melibiose, and raffinose had a potential effect to increase the PI size of the larvae significantly (*p* < 0.01–0.0001). The results are shown in Figure 6.

### 3.5. The Investigation of the Obesity-Preventing Effect of the D. kaki Fruit Extract and Its Oligosaccharides

Zebrafish is an efficient tool for the rapid identification of cellular morphology and genetic studies. High-fat-diet induction increased whole-body adipocytes of zebrafish larvae by egg yolk overfeeding [25]. To hasten natural product research and development for metabolic syndrome, zebrafish is a predictive and convenient tool with a low cost and a short period of drug screening [26]. *D. kaki* and its oligosaccharides were examined by co-treatment to investigate their obesity-preventing effects. A total of 9 dpf of zebrafish larvae were imaged and determined the belly size by measuring the abdominal length. The overfeeding of a high-fat diet increased the size of the zebrafish bodies as evidenced by the significant increase in abdominal length (*p* < 0.0001), which was prevented in the treatment groups. The *D. kaki* extract and oligosaccharide exposure groups showed a significantly decreased size of zebrafish abdominal length, *p* < 0.01–0.0001 (Figure 7A). Lipid accumulation was monitored by the intensity histogram of Nile red staining fluorescence image. The obesity group increased the intensity in the abdominal area significantly (*p* < 0.0001) and the treatment groups lowered the lipid accumulation by decreasing the intensity, as shown in Figure 7B (*p* < 0.001–0.0001). As a result of this experiment, *D. kaki* and its oligosaccharides exhibited an obesity-preventing effect in the zebrafish model via lowering the lipid accumulation in the larvae body.

### 3.6. Genes Expression on High-Fat-Diet-Induced Obesity in Zebrafish Larvae

The mechanism studies of high-fat-diet-induced obesity in zebrafish larvae were performed via RT-qPCR to evaluate the expression of inflammatory, lipogenesis, and lipid-lowering-related gene. The inflammatory-related mRNA tumor necrosis factor alpha (*TNF-α*), interleukin 6 (*IL-6*), and interleukin *1β* (*IL-1β*) were increased significantly in the obesity group. Gentiobiose and melibiose reduced the expression of three of the inflammatory-related genes. *TNF-α* and *IL-6* were reduced significantly in the *D. kaki* fruit extract treatment group. The results show that lipogenesis-related fatty acid synthase (*FASN*) gene expression significantly increased by all treatments, while the *D. kaki* fruit extract, gentiobiose, and melibiose reduced sterol regulatory element binding transcription factor 1 (*SREBF1*). Lipid-lowering-related gene, carnitine palmitoyltransferase 1A (*CPT1A*), was reduced by the *D. kaki* fruit extract and gentiobiose significantly. The results are shown in Figure 8.

### 3.7. Enzyme Activity Assays of PTP1B to D. kaki Fruit and Oligosaccharides

The inhibition of PTP1B was performed to investigate in vitro activity. *D. kaki* showed PTP1B inhibition as the IC_50_ value of 693.3 µg/mL. In contrast, at the concentration of 10 mg/mL, none of the oligosaccharides showed an inhibitory effect against PTP1B. From the results of molecular docking, the three oligosaccharides gentiobiose, melibiose, and raffinose interacted with the catalytic sites of PTP1B, but were inactive in the enzyme assay. The active catalytic site of PTP1B was the phosphotyrosine (pTyr)-binding pocket, which is located in the intracellular region [27]. Therefore, the cellular penetration of the cell membrane was necessary for the PTP1B inhibitors [28]. On the other hand, the allosteric site was less conserved and may enhance the selectivity and bioavailability of the inhibitors. This may explain why the function of the catalytic site of PTP1B had a poor inhibitory effect in vitro study [29].

## 4. Conclusions

*D. kaki* fruit extract, gentiobiose, melibiose, and raffinose showed anti-insulin resistance and obesity-preventing effects by protecting β cells from high doses of human insulin induction, decreasing the abdominal size, and reducing the lipid accumulation in obese zebrafish. The mechanism of action of the oligosaccharides was studied via molecular docking and enzyme activities of PTP1B. Three of the oligosaccharides showed a binding interaction with the active catalytic sites of PTP1B, but did not show an inhibition effect in the enzyme assay. This result confirmed that the catalytic sites of PTP1B are highly conserved structures from bacteria to mammals [30], and cellular penetration was essential for the inhibitors [28]. In addition, the large uncharged polar molecule of oligosaccharides cannot diffuse through the cell membrane [31]; this may be the reason of enzyme activity results. On the other hand, there is evidence for the intestinal transportation of non-digestible oligosaccharides in clinical studies. Non-digestible oligosaccharides might directly contact with systemic immune cell enter through the gut barrier when the gut permeability was increased, such as in inflammatory bowel disease (IBD), obesity, type 1 diabetes, and non-alcoholic fatty liver disease [32].

Previous literature studies have shown that raffinose and melibiose promoted the growth of bifidobacteria in overweight children. Due to their non-digestibility, these oligosaccharides may have beneficial effects in the obese patient’s colon [33]. Moreover, raffinose was reported to inhibit lipid accumulation in HepG2 and 3T3-L1 cells and increase glucose uptake in vivo study [34]. To apply the effects of *D. kaki* and oligosaccharides on the metabolic syndrome to humans, studies of metabolic-related gene expression are required. The pathological characteristics of the inflammation pathway (*IL-6*, *IL-1β*, and apolipoprotein H (*APOH*)) and lipid metabolism (*SREBF1*), peroxisome proliferator-activated receptors α/γ (*PPARα/γ*), nuclear receptor subfamily 1 group H member 3 (*NR1H3*), and leptin (*LEP*)) between human and zebrafish have been divulged [35]. Thus, further studies on the role of the mechanism of function of these oligosaccharides will contribute to the development of dietary supplements.

From our studies, the *D. kaki* fruit extract and oligosaccharides resulted in gene expression changing inflammation, lipogenesis, and lipid-lowering-related genes. The fruit extract decreased the expression of *TNF-α* and *IL-6*, which impair the insulin action in metabolic tissues [36], and *SREBF1* and *FASN*, which regulate lipogenesis [37], and *CPT1A*, which promote the fatty acid metabolism by improving lipid levels and glucose [38]. The oligosaccharide gentiobiose downregulated the proinflammatory cytokines *TNF-α*, *IL-6*, *IL-1β* (interleukin 1*β* is the production of *β*-cell oxidative stress that promote macrophages activation [39]), lipogenesis-related *SREBF1*, *FASN*, and lipid-lowering-related *CPT1A.* Moreover, the regulation of *TNF-α*, *IL-6*, *IL-1β*, *SREBF1*, and *FASN* expression was decreased by melibiose. In contrast, raffinose only downregulated *FASN* expression. In conclusion, gentiobiose, melibiose, and the *D. kaki* fruit extract have a great potential for applications in metabolic syndrome drug development.

## Figures and Tables

**Figure 1 nutrients-14-03249-f001:**
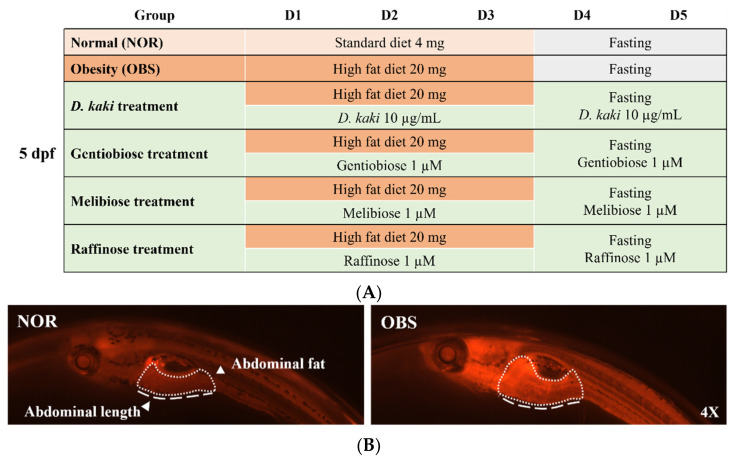
Zebrafish as an obesity model. Experimental setup (**A**). Comparison of the abdominal length and the abdominal fat between the normal group (NOR) and obesity group (OBS) stained with Nile red staining and observed under florescence microscope at day five of the experiment (**B**).

**Figure 2 nutrients-14-03249-f002:**
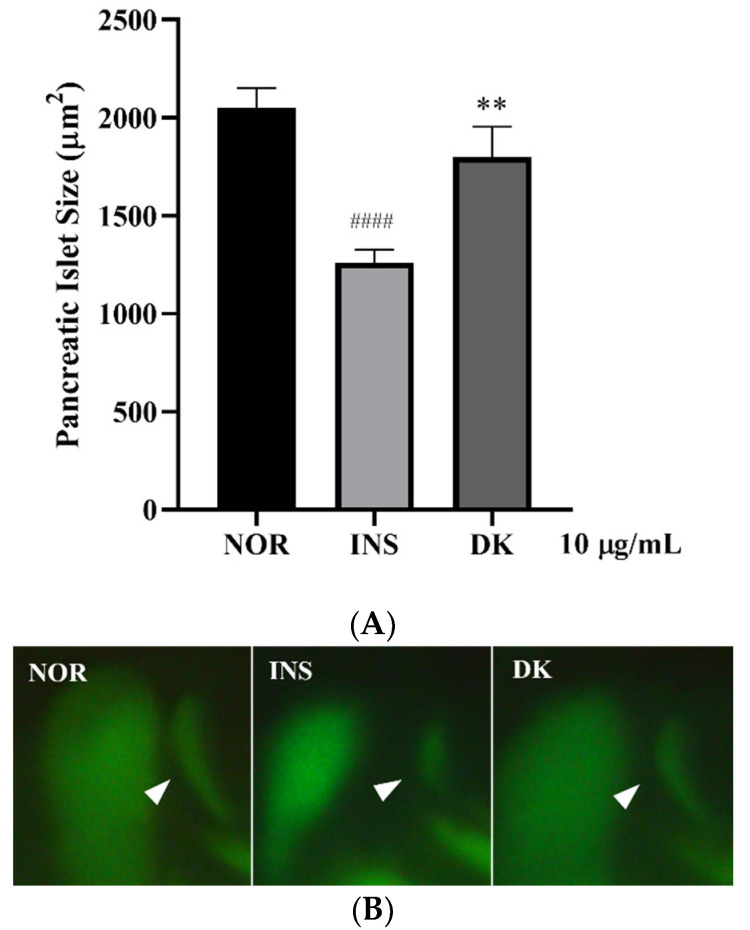
Screening of the *D. kaki* fruit extract for anti-insulin resistance activity in the zebrafish larvae. The treatment group with the *D. kaki* fruit extract significantly recovered, as shown by the PI size (**A**). Image of PI (pancreatic islet) staining by 40 µM 2-NBDG (**B**), normal (NOR), insulin induction (INS), and *D. kaki* (DK). #### *p* < 0.0001 compared to the control (NOR). ** *p* < 0.01 compared to insulin induction (INS). The statistical significance was determined using a one-way ANOVA with Dunnett’s post hoc test, expressed as means ± SEM (*n* = 20).

**Figure 3 nutrients-14-03249-f003:**
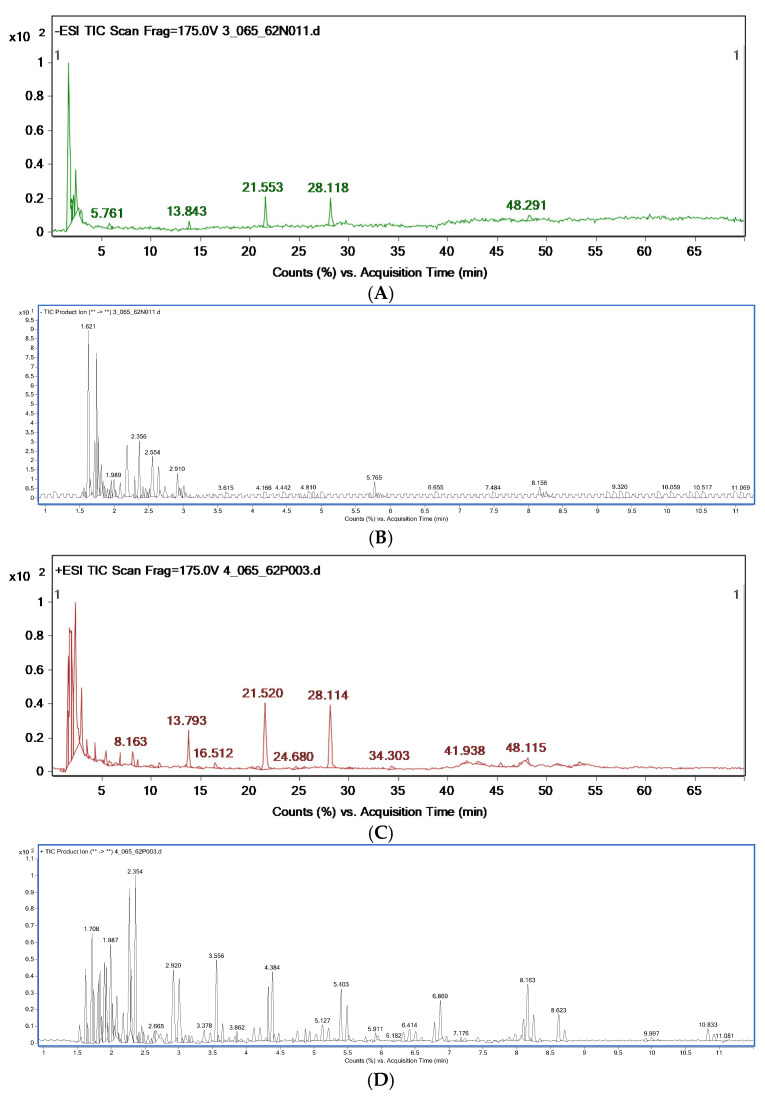
LC-ESI-MS/MS (**A**) and the expanded chromatogram created by plotting the total ion current in a series of product ions recorded as a function of the retention time of LC-ESI-MS/MS (**B**). LC+ESI-MS/MS (**C**) chromatograms of the *D. kaki* water extract and the expanded chromatogram created by plotting the total ion current in a series of product ions recorded as a function of the retention time of LC+ESI-MS/MS (**D**).

**Figure 4 nutrients-14-03249-f004:**
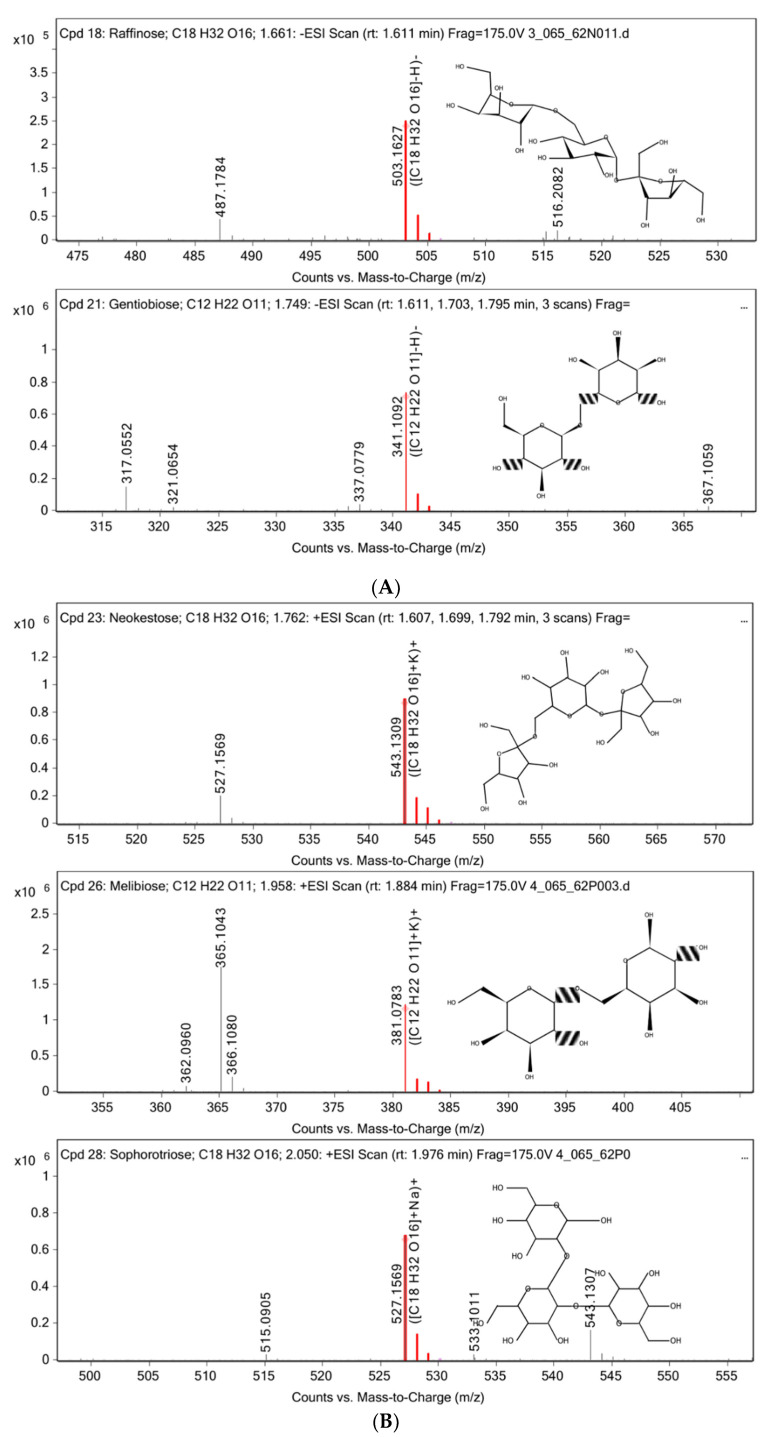
Mass spectrum of raffinose and gentiobiose from −ESI mode (**A**). Neokestose, melibiose, and sophorotriose from +ESI mode (**B**).

**Figure 5 nutrients-14-03249-f005:**
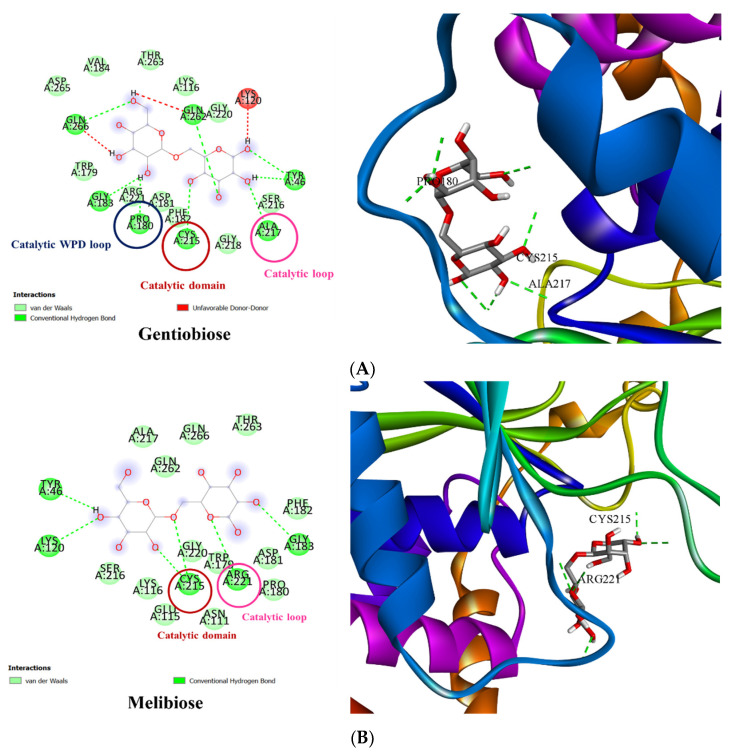
The crystallographic structure of PTP1B molecular docking related five oligosaccharides. Hydrogen bond of gentiobiose binding to the catalytic active sites of PTP1B Pro180, Cys215, and Ala217 residue of the catalytic WPD loop, catalytic domain, and catalytic loop, respectively (**A**). Melibiose showed ligand binding to Cys215 of the catalytic domain and Arg221 of the catalytic loop (**B**). Pro180 and Cys215 of the catalytic WPD loop and catalytic domain were bound to raffinose with hydrogen bonding (**C**). Neokestose and sophorotriose were not bound to any of the active residues (**D**).

**Figure 6 nutrients-14-03249-f006:**
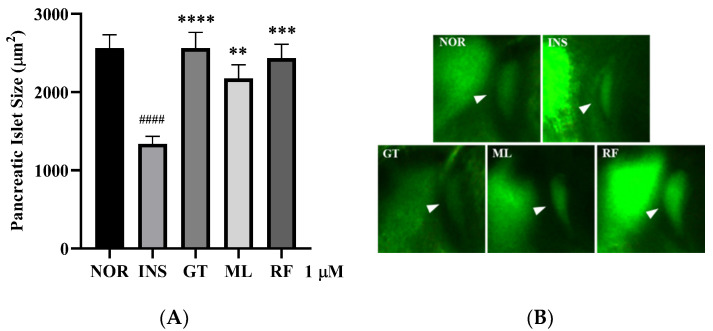
Oligosaccharides from *D. kaki* significantly showed the recovery of β cells by decreasing the size of zebrafish PI (**A**). The images of 2-NBDG staining of PI (**B**), normal (NOR), insulin induction (INS), gentiobiose (GT), melibiose (MB), and raffinose (RF). #### *p* < 0.0001 compared to NOR. **** *p* < 0.0001, *** *p* < 0.001, ** *p* < 0.01 compared to INS. The statistical significance was determined using a one-way ANOVA with Dunnett’s post hoc test, expressed as means ± SEM (*n* = 20).

**Figure 7 nutrients-14-03249-f007:**
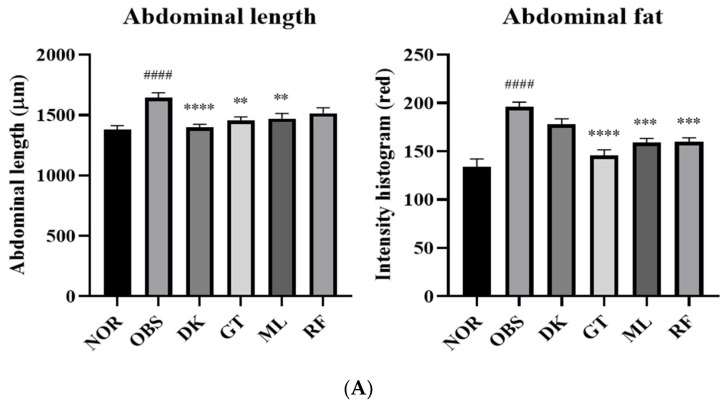
A total of 10 µg/mL of *D. kaki* (DK) and 1 µM of the oligosaccharides significantly decreased the abdominal size of and lipid accumulation in obese zebrafish by co-treatment (**A**). The images of 0.5 µg/mL Nile red staining of zebrafish body (**B**), normal (NOR), obesity induction (OBS), *D. kaki* (DK), gentiobiose (GT), melibiose (MB), and raffinose (RF). #### *p* < 0.0001 compared to NOR. **** *p* < 0.0001, *** *p* < 0.001, ** *p* < 0.01 compared to OBS. The statistical significance was determined using a one-way ANOVA with Dunnett’s post hoc test, expressed as means ± SEM (*n* = 20).

**Figure 8 nutrients-14-03249-f008:**
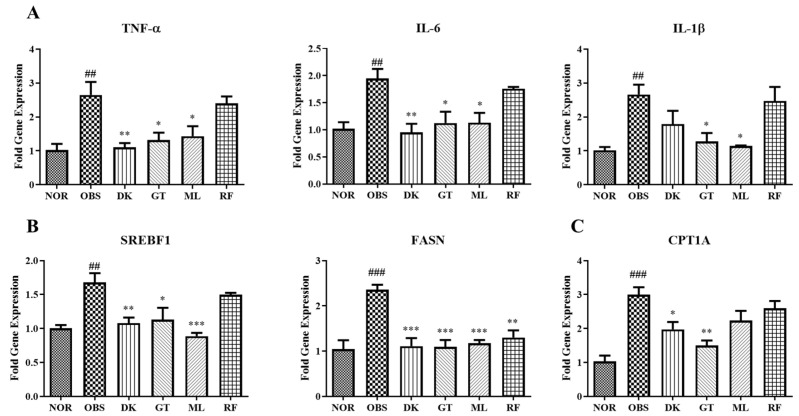
Gene expression on high-fat-diet-induced obesity in zebrafish larvae. The results of RT-qPCR evaluation on inflammatory-related genes *TNF-α*, *IL-6*, and *IL-1β* (**A**). The results of RT-qPCR evaluation on the lipogenesis-related genes *SREBF1* and *FASN* (**B**). The results of RT-qPCR evaluation on lipid-lowering-related genes *CPT1A* (**C**). Normal (NOR), obesity induction (OBS), *D. kaki* (DK), gentiobiose (GT), melibiose (MB), and raffinose (RF). ### *p* < 0.001, ## *p* < 0.01 compared to NOR. *** *p* < 0.001, ** *p* < 0.01, * *p* < 0.05 compared to OBS. The statistical significance was determined using a one-way ANOVA with Dunnett’s post hoc test, expressed as means ± SEM (*n* = 20).

**Table 1 nutrients-14-03249-t001:** Primers for RT-qPCR.

Gene	Forward Sequence	Reverse Sequence
TNF-α	GCT TAT GAG CCA TGC AGT GA	TGC CCA GTC TGT CTC CTT CT
IL6	AAG GGG TCA GGA TCA GCA C	GCT GTA GAT TCG CGT TAG ACA TC
IL1B	TGG CGA ACG TCA TCC AAG	GGA GCA CTG GGC GAC GCA TA
SREBF1	CAT CCA CAT GGC TCT GAG TG	CTC ATC CAC AAA GAA GCG GT
FASN	GAG AAA GCT TGC CAA ACA GG	GAG GGT CTT GCA GGA GAC AG
CPT1A	CAT GAG GCT CTT CGG CAA	AAG AGC AGG CCT AAG GAT G

**Table 2 nutrients-14-03249-t002:** Chemical constituents of the *D. kaki* extract detected in LC-ESI-MS/MS and LC+ESI-MS/MS chromatograms.

RT	Height	Mass	Identification Result
LC-ESI-MS/MS chromatograms			
1.661	497,154	504.17	Raffinose
1.749	4,200,859	342.1166	Gentiobiose
1.991	482,479	342.1165	Gentiobiose
2.188	1,329,997	646.1966	beta-D-4-Deoxy-delta4-GlcA-(1->4)-beta-D-Glc-(1->4)-alpha-L-Rha-(1->3)-beta-D-Glc
2.318	294,628	342.1169	Gentiobiose
2.33	73,904	192.0268	2,5-Didehydro-D-gluconate
2.464	252,822	219.1112	Pantothenic Acid
2.602	1,051,653	224.0327	Dehydrochorismic acid
2.811	145,798	342.1157	D-(+)-Cellobiose
5.836	415,037	382.1839	1,2,10-Trihydroxydihydro-trans-linalyl oxide 7-O-beta-D-glucopyranoside
46.14	208,962	294.1834	Myrsinone
48.25	409,723	328.2251	(9R,10S,12Z)-9,10-Dihydroxy-8-oxo-12-octadecenoic acid
LC+ESI-MS/MS chromatograms			
1.59	142,819	317.1212	N-(1-Deoxy-1-fructosyl) histidine
1.762	4,633,119	504.1678	Neokestose
1.933	6,561,444	261.1204	Epidermin
1.958	1,734,862	342.1152	Melibiose
2.05	1,517,744	504.1676	Sophorotriose
2.234	735,399	438.1151	Loquatoside
2.301	6,245,852	271.1647	Prolyl-Arginine
2.519	998,209	342.1153	Sucrose
2.703	1,104,006	366.1413	N-(1-Deoxy-1-fructosyl) tryptophan
3.013	6,275,606	187.063	3-Amino-2-naphthoic acid
3.486	1,147,920	422.2117	6alpha-Fluoro-17-hydroxycorticosterone 21-acetate
3.615	266,556	418.1823	Glu Arg Asp
3.841	595,699	478.1672	Kelampayoside A
4.406	758,433	444.2559	Arg Asn Arg
4.912	1,230,160	361.233	Lys Lys Ser
5.13	1,574,844	510.2643	Nebramycin factor 5′
5.498	4,669,055	436.2273	Flurandrenolide
5.82	568,261	404.1644	Asp Asp Arg
6.418	1,185,915	532.3082	5β-Cyprinolsulfate
6.879	3,665,061	458.2718	Arg Gln Arg
8.719	2,455,730	486.3239	Docosahexaenoyl Serotonin
10.941	1,224,227	568.3056	Ceanothine E
16.685	162,831	538.2945	Euphorbia factor Ti2
21.609	2,415,031	672.4863	PE-Cer(d14:2(4E,6E)/20:1(11Z)(2OH))
28.06	5,609,787	775.6128	PE(21:0/17:0)
42.038	1,513,417	273.2659	C16 Sphinganine
43.05	1,422,944	229.24	Xestoaminol C
48.205	1,502,361	317.2921	Phytosphingosine

## Data Availability

Not applicable.

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
