# Peer review of "The Effects of Persimmon (Diospyros kaki L.f.) Oligosaccharides on Features of the Metabolic Syndrome in Zebrafish"

_nutrients, 2022, doi:10.3390/nu14163249_

Round 1

Reviewer 1 Report

In the manuscript “The Effects of Oligosaccharides from Persimmon (Diospyros kaki L.f.) on Metabolic Syndrome, and its impacts to PTP1B” Nuankaew et coworkers investigated the effects of Persimmon oligosaccharides on insulin resistance and obesity in Zebrafish, as well as evaluating its activity on Protein Tyrosine Phosphatase 1B (PTP1B).

In my opinion, this paper is too simplistic and it is full of inaccuracies as well as wrong and misleading statements (i.e. page 1, lines 30-33).

In addition, most of the Authors’ claims are lacking of appropriate references (i.e. page 1, lines 37 38 and page 2, lines 48-49).

Moreover, a more in-depth evaluation of mechanisms underlining Persimmon effects are needed (i.e., gene and protein expression analyses).

In page 4, line 141, Authors write that “A p value > 0.05 (∗) was considered statistically significant”, however p value should be below 0.05 to consider outcomes significant.

Images provided are not clear and almost confounding.

English level is very poor, thus extensive editing of English language and style are absolutely required.

Author Response

Dear reviewer,

On behalf of my coauthors, I would like to thank you for the opportunity to revise and resubmit our manuscript nutrients-1720490, entitled “The Effects of Oligosaccharides from Persimmon (Diospyros kaki L.f.) on Metabolic Syndrome, and its impacts to PTP1B”

We found the reviewers’ comments to be helpful in revising the manuscript and have carefully considered and responded to each suggestion. In the majority of cases we were successful in incorporating the reviewers’ feedback into our revised manuscript.

There was one suggestion that we were unable to implement because the of revision deadline that can not get the data of gene expression, if the reviewers and the editors think pcr was necessary, we will prepare immediately.

The manuscript was checked from a native English speaker with scientific expertise.

Thank you again for your consideration of our revised manuscript. 

Best regards

Wanlapa Nunkaew

Reviewer 2 Report

Metabolic syndrome is a complex disease with hypertension, insulin/glucose intolerance leading to diabetes, fatty liver, altered blood lipids and obesity leading to a range of cardiovascular, metabolic and other disorders. Many biochemical and physiological systems are involved. It is a vast oversimplification to use just one parameter (lipid accumulation) and one mechanism of action (PTP1B inhibition which did not occur with pure oligosaccharides) in one animal species (zebrafish) to define the human syndrome and suggest options for therapeutic intervention. Zebrafish may be high efficiency but low relevance to this human disease state.

Mass spectroscopic analysis defines compounds that are present, but not the amount – this is a critical parameter in pharmacology! As such, please define concentrations of polysaccharides used in this study, for comparison with literature studies.

Please include in the Discussion previous literature studies showing that gentiobiose, melibiose and raffinose have biological responses that are relevant to the study of obesity.

Section 3.6: The explanation of why pure oligosaccharides did not inhibit PTP1B looks like an admission that the wrong compounds were chosen for the study. This lack of activity also demonstrates that pharmacokinetics is critical – if a compound does not get to the site of action, then there will be no physiological or biochemical response. Also, the authors have used a computer simulation – maybe this simulation does not mimic real life? My conclusion from these results is that the chosen oligosaccharides simply have no activity (in addition to no access to enzyme).

The authors need assistance from a native English speaker with scientific expertise to make the manuscript more understandable.

Author Response

Dear reviewer,

On behalf of my coauthors, I would like to thank you for the opportunity to revise and resubmit our manuscript nutrients-1720490, entitled “The Effects of Oligosaccharides from Persimmon (Diospyros kaki L.f.) on Metabolic Syndrome, and its impacts to PTP1B”

We found the reviewers’ comments to be helpful in revising the manuscript and have carefully considered and responded to each suggestion. In the majority of cases we were successful in incorporating the reviewers’ feedback into our revised manuscript.

The manuscript was checked from a native English speaker with scientific expertise.

Thank you again for your consideration of our revised manuscript. 

Best regards

Wanlapa Nunkaew

Reviewer 3 Report

Some studies have shown that zebrafish may be a model system for studying metabolic diseases because of the functional conservation in lipid metabolism, adipose biology, pancreas structure, and glucose homeostasis. However, the physiology and metabolism of zebrafish are obviously different from those of human.

This is an interesting study. There are, however, several questions need to be answered and clarified in this manuscript.

Major concern:

1.     The authors should explain why zebrafish but not rat was used as animal model in this study.

2.     The authors should explain how the results from this study apply to human.

3.     The title is “….on metabolic syndrome….”.  Please describe how the fatty zebrafish show metabolic syndrome.

4.     Please describe some related literatures concerning zebrafish is a metabolic syndrome model.

Author Response

Dear reviewer,

On behalf of my coauthors, I would like to thank you for the opportunity to revise and resubmit our manuscript nutrients-1720490, entitled “The Effects of Oligosaccharides from Persimmon (Diospyros kaki L.f.) on Metabolic Syndrome, and its impacts to PTP1B”

We found the reviewers’ comments to be helpful in revising the manuscript and have carefully considered and responded to each suggestion. In the majority of cases we were successful in incorporating the reviewers’ feedback into our revised manuscript.

Thank you again for your consideration of our revised manuscript. 

Best regards

Wanlapa Nunkaew

Round 2

Reviewer 1 Report

Authors did not adequately answer most of the points, so I recommend answering each point and if not possible, explaining why in the cover letter.

In addition, as I mentioned above, a more thorough evaluation of the mechanisms underlying Persimmon's effects is needed, so gene and/or protein expression analyses are absolutely required.

Author Response

Dear reviewer,

On behalf of my coauthors, I would like to thank you for the opportunity to revise and resubmit our manuscript nutrients-1720490, entitled “The Effects of Oligosaccharides from Persimmon (Diospyros kaki L.f.) on Metabolic Syndrome, and its impacts to PTP1B in Zebrafish”

We found the reviewers’ comments to be helpful in revising the manuscript and have carefully considered and responded to each suggestion. In the majority of cases, we were successful in incorporating the reviewers’ feedback into our revised manuscript.

The manuscript was checked from a native English speaker with scientific expertise.

Thank you again for your consideration of our revised manuscript.

Best regards

Wanlapa Nunkaew

  1. Wrong and misleading statements (i.e. page 1, lines 30-33).
  • Statements were changed to “The consumption of quality fats and carbohydrates is pivotal to improve blood lipids and glycemic index in metabolic syndrome patient [2].” (page 1, lines 33-34)

  1. Inappropriate references in page 1, lines 37 38 and page 2, lines 48-49.
  • The references were appropriately divided in page 1, lines 42, 43 and page 2, lines 57, 58.

  1. More in-depth evaluation of mechanisms underlining Persimmon effects are needed.
  • Gene and protein expression analyses were added (page 4, 13-15).

2.8. Real Time-quantitative Polymerase Chain Reaction (RT-qPCR)

Zebrafish larvae were fed with the high fat diet for three days then treated by D. kaki fruit extract and the oligosaccharides for another two days. The extraction of total RNA using TRIzolTM reagent (Thermo Fisher Scientific, Korea). Reverse transcription was performed by Reverse Aid First Strand cDNA Synthesis Kit (Thermo Fisher Scientific) for the synthesis of cDNA. The qPCR was performed in 10 μL reactions containing 5 μL of SYBR Select Master Mix (Applied Biosystems, Thermo Fisher Scientific), 1 μL of cDNA template, 1 μL of forward primer (10 pmol), 1 μL of reverse primer (10 pmol), and 2 μL of RNase free water. The qPCR parameters were initial denaturation at 95°C for 5 min, followed by 45 cycles of 95°C for 15 s, 60°C for 15 s and 72°C for 20 s, and then 73°C for 5 min. Primer sequences are listed in Table 1. The expression of genes was analyzed by the 2−ΔΔCt method.

Table 1. Primers for RT-qPCR

Gene

Forward Sequence

Reverse Sequence

TNF-α

GCT TAT GAG CCA TGC AGT GA

TGC CCA GTC TGT CTC CTT CT

IL6

AAG GGG TCA GGA TCA GCA C

GCT GTA GAT TCG CGT TAG ACA TC

IL1B

TGG CGA ACG TCA TCC AAG

GGA GCA CTG GGC GAC GCA TA

SREBF1

CAT CCA CAT GGC TCT GAG TG

CTC ATC CAC AAA GAA GCG GT

FASN

GAG AAA GCT TGC CAA ACA GG

GAG GGT CTT GCA GGA GAC AG

CPT1A

CAT GAG GCT CTT CGG CAA

AAG AGC AGG CCT AAG GAT G

3.6. Genes expression on High Fat Diet-Induced Obesity in Zebrafish Larvae

The mechanism studies of high fat diet-induced obesity in zebrafish larvae were performed via RT-qPCR experiment to evaluate the gene expression on inflammation, lipogenesis, and lipid-lowering. The inflammation-related mRNA TNF-α, IL-6, and IL-1β were increased significantly in the obesity group. Gentiobiose reduced the expression of three of the inflammatory genes. TNF-α and IL-6 were reduced significantly in D. kaki fruit extract treatment group. Melibiose reduced IL-6 and IL-1β significantly. From the results showed that all of drugs treatment groups were increased significantly in lipogenesis-related FASN gene but only D. kaki fruit extract and melibiose reduced SREBF1. Lipid-lowering related gene CPT1A was reduced by D. kaki fruit extract and gentiobiose significantly. The results were shown in Figure. 8.

Figure 8. Genes expression on High Fat Diet-Induced Obesity in Zebrafish Larvae. The results of RT-qPCR evaluation on inflammation-related genes TNF-α, IL-6, and IL-1β (A). The results of RT-qPCR evaluation on lipogenesis-related genes SREBF1 and FASN (B). The results of RT-qPCR evaluation on lipid-lowering related genes CPT1A (C). Normal (NOR), obesity induction (OBS), D. kaki (DK), gentiobiose (GT), melibiose (MB), and raffinose (RF). ## p < 0.01, # p < 0.05 compared to NOR. ∗∗∗ p < 0.001, ∗∗ p < 0.01, ∗ p < 0.05 compared to OBS. The statistical significance was determined using paired t-test, expressed as means ± SEM.

From our studies, D. kaki fruit extract and the oligosaccharides resulted in gene expression changes in inflammation, lipogenesis, and lipid-lowering related genes. The fruit extract has decreased the expression of tumor necrosis factor alpha (TNF-α) and interleukin 6 (IL-6) which impair the insulin action in metabolic tissues [34], sterol regulatory element binding transcription factor 1 (SREBF1) and fatty acid synthase (FASN) which regulate the lipogenesis [35], and carnitine palmitoyltransferase 1A (CPT1A) which promote the fatty acid metabolism by improving lipid levels and glucose [36]. The oligosaccharide gentiobiose downregulated the proinflammatory cytokines TNF-α, IL-6, IL-1β (interleukin 1β is the production of β cell oxidative stress that promote macrophages activation [37]), lipogenesis related FASN and lipid-lowering related CPT1A. Moreover, the regulation of IL-6, IL-1β, SREBF1, and FASN expression were decreased by melibiose. In contrast, raffinose only downregulated FASN expression. In conclude, gentiobiose, melibiose, and D. kaki fruit extract have a great potential for applications to metabolic syndrome drug development.

  1. In page 4, line 141, Authors write that “A p value > 0.05 (∗) was considered statistically significant”
  • page 4, line 172 was changed to “p value <05 (∗) …”

  1. Images provided are not clear and almost confounding.
  • The images were changed in page 10-11

  1. English level is very poor, thus extensive editing of English language and style are absolutely required.
  • The manuscript had done the grammar check.

Reviewer 2 Report

1. Lines 56-57: Please provide reference for this statement (and use plural so "oligosaccharides").

2. Line 58: This sentence does not make sense. 

3. Line 85: verb needed in sentence.

4. Legend to figure 2: Please define "PI". 

5. The text still needs careful correction for English usage by a native speaker with scientific expertise. 

Author Response

Dear reviewer,

On behalf of my coauthors, I would like to thank you for the opportunity to revise and resubmit our manuscript nutrients-1720490, entitled “The Effects of Oligosaccharides from Persimmon (Diospyros kaki L.f.) on Metabolic Syndrome, and its impacts to PTP1B in Zebrafish”

We found the reviewers’ comments to be helpful in revising the manuscript and have carefully considered and responded to each suggestion. In the majority of cases, we were successful in incorporating the reviewers’ feedback into our revised manuscript.

The manuscript was checked from a native English speaker with scientific expertise.

Thank you again for your consideration of our revised manuscript.

Best regards

Wanlapa Nunkaew

  1. Lines 56-57: Please provide reference for this statement (and use plural so "oligosaccharides").
  • Line 60: [18] Zhu, D.; Yan, Q.; Liu, J.; Wu, X.; Jiang, Z. Can functional oligosaccharides reduce the risk of diabetes mellitus?. FASEB J. 2019, 33(11), 11655–11667.

  1. Line 58: This sentence does not make sense. 
  • Line 60-63: Referring to the glycosidic bond between monomeric sugar units of the oligosaccharides cannot be broken by human gastrointestinal digestive enzymes, this characteristic provided nondigestible and nonsusceptible in the human digestive system [19].

  1. Line 85: verb needed in sentence.
  • Line 90: Mass spectra were analyzed by MS/MS spectral mass library.

  1. Legend to figure 2: Please define "PI". 
  • Line 184: “Image of PI (pancreatic islet)”

  1. The text still needs careful correction for English usage by a native speaker with scientific expertise. 
  • The manuscript had done the grammar check.

Reviewer 3 Report

The title of this manuscript should be changed into “ The Effects of Oligosaccharides from Persimmon (Diospyros 2 kaki L.f.) on Metabolic Syndrome, and its impacts to PTP1B in Zebrafish” because the animal model is zebrafish.

Author Response

Dear reviewer,

On behalf of my coauthors, I would like to thank you for the opportunity to revise and resubmit our manuscript nutrients-1720490, entitled “The Effects of Oligosaccharides from Persimmon (Diospyros kaki L.f.) on Metabolic Syndrome, and its impacts to PTP1B in Zebrafish”

We found the reviewers’ comments to be helpful in revising the manuscript and have carefully considered and responded to each suggestion. In the majority of cases, we were successful in incorporating the reviewers’ feedback into our revised manuscript.

The title was changed to “The Effects of Oligosaccharides from Persimmon (Diospyros kaki f.) on Metabolic Syndrome, and its impacts to PTP1B in Zebrafish”

The manuscript was checked from a native English speaker with scientific expertise.

Thank you again for your consideration of our revised manuscript.

Best regards

Wanlapa Nunkaew
